# Social Support: The Effect on Nocturnal Blood Pressure Dipping

**DOI:** 10.3390/ijerph20054579

**Published:** 2023-03-04

**Authors:** Wendy C. Birmingham, Anna Jorgensen, Sinclaire Hancock, Lori L. Wadsworth, Man Hung

**Affiliations:** 1Psychology Department, Brigham Young University, Provo, UT 84602, USA; 2Romney Institute of Public Management, Brigham Young University, Provo, UT 84602, USA; 3Department of Orthopedic Surgery Operations, University of Utah, 590 Wakara Way, Salt Lake City, UT 84108, USA; 4College of Dental Medicine, Roseman University of Health Sciences, 10894 S River Front Pkwy, South Jordan, UT 84095, USA; 5George E. Wahlen Department of Veteran Affairs Medical Center, 500 Foothill Drive, Salt Lake City, UT 84148, USA

**Keywords:** blood pressure, nocturnal dipping, social support

## Abstract

Social support has long been associated with cardiovascular disease risk assessed with blood pressure (BP). BP exhibits a circadian rhythm in which BP should dip between 10 and 15% overnight. Blunted nocturnal dipping (non-dipping) is a predictor of cardiovascular morbidity and mortality independent of clinical BP and is a better predictor of cardiovascular disease risk than either daytime or nighttime BP. However, it is often examined in hypertensive individuals and less often in normotensive individuals. Those under age 50 are at increased risk for having lower social support. This study examined social support and nocturnal dipping in normotensive individuals under age 50 using ambulatory blood pressure monitoring (ABP). ABP was collected in 179 participants throughout a 24-h period. Participants completed the Interpersonal Support Evaluation List, which assesses perceived levels of social support in one’s network. Participants with low levels of social support demonstrated blunted dipping. This effect was moderated by sex, with women showing greater benefit from their social support. These findings demonstrate the impact social support can have on cardiovascular health, exhibited through blunted dipping, and are particularly important as the study was conducted in normotensive individuals who are less likely to have high levels of social support.

## 1. Introduction

Social support can be defined as the perceived availability of resources [1] (functional support). Research has shown a strong association between social support and mortality and morbidity [2,3,4,5], including cardiometabolic diseases such as cardiovascular disease (CVD), type 1 and type 2 diabetes, and chronic obstructive disease, which are the global leading causes of death [6,7,8,9,10]. Lower levels of social support have been associated with higher incidence and progression of colorectal cancer in men, higher recurrence of breast cancer in women, worse outcomes in older adults with lung cancer, worse outcomes for single individuals with gastric cancer [11,12,13,14], and worse diabetes outcomes [15]. Social support has also been linked to psychological factors such as depression, distress, and satisfaction with life, which have influences on cardiovascular health [16,17]. A review by Holt-Lunstad, Smith & Layton [18] found that the link between supportive relationships and health was as predictive of disease as known risk factors such as smoking and lack of physical exercise. Additionally, individuals with poor social connectedness are 29% more likely to develop CVD and 32% more at risk for stroke [19].

Hypertension is the most common disease in industrialized nations [20] and is the predominant risk factor for CVD [21]. In 2020, more than 670,000 deaths in the U.S. had hypertension as a primary or contributing cause [20], and nearly half of adults in the U.S. (47%) have hypertension, as defined by the American Heart Association [22]. In adults who have not been diagnosed with CVD, there is a strong association of slightly elevated levels of both systolic blood pressure (SBP) and diastolic blood pressure (DBP) with increased risk for developing hypertension in a relatively short time [23] and is associated with early target organ damage [24,25,26]. This is important, as blood pressure (BP) shows an increased trajectory over time and is associated with poor cardiovascular outcomes even 25 years later [27]. Thus, several meta-analyses have shown the effectiveness of lowering BP to reduce CVD risk [28,29].

BP shows a circadian rhythm, such that a healthy cardiovascular profile includes a decrease of 10–20% from day to night (i.e., nocturnal dipping) [30]. Blunted nocturnal dipping (non-dipping) is defined as blood pressure which does not dip at least 10%. It is associated with increased risk for cardiovascular events in both normotensive and hypertensive adults, higher risk of cardiovascular morbidity and mortality [31,32,33], composite kidney endpoint, and increased risk of all-cause mortality [34]. Recent research has shown blunted nocturnal dipping to be a better predictor of cardiovascular disease and mortality than 24-h averages alone [35,36]. Indeed, abnormalities of the circadian dipping patterns are associated with both total and cardiovascular mortality [36].

Ambulatory blood pressure (ABP) has been demonstrated to be a better predictor of mortality than blood pressure taken in an office setting, typically a physician’s office (clinical BP) [37,38,39]. ABP allows for a more accurate assessment of BP, as it takes multiple readings spread throughout intervals across the day and night and can thus provide a more accurate evaluation of BP fluctuations, rather than a single reading in a physician’s office, which may be influenced by the white-coat effect [38]. However, while hypertension is the primary predictor of CVD and has been consistently linked to social support which has been assessed with ABP [40,41,42,43], a recent meta-analysis by Uchino and colleagues [44] found no association between daytime ABP and social support. This is an interesting finding given the amount of literature detailing the association between the two. For nocturnal dipping research, these are important findings to take into consideration, as nocturnal dipping is calculated using daytime and nighttime ABP. It would be worth considering that social support may differentially impact ABP when using the ratio of daytime and nighttime dipping ABP rather than just daytime ABP. Uchino and colleagues suggested that an examination of other indices of ABP such as nocturnal dipping would be beneficial, as such studies were not included in the meta-analysis. This could be informative given that nocturnal dipping as assessed with ABP has been linked to social support, such that those with lower levels of social support/social integration showed blunted dipping [45,46,47].

However, much of the research on nocturnal dipping has focused more heavily on hypertensive individuals and older adults, or it has not differentiated between younger and older adults. Yet adults under the age of 50 tend to report a lack of social connections or more loneliness than those over the age of 50 [48,49]. In fact, younger generations (Gen Z and Millennials) report a lack of social support and fewer social interactions than baby boomers [49]. Further, the literature on dipping has varied, with some studies showing social support impacting dipping on both SBP and DBP, while other studies show dipping on only one or the other (either SBP or DBP), or with no significant effects on either. A 2013 meta-analysis by Fortmann and Gallo [50] showed that of the studies that used the Interpersonal Support Evaluation List (ISEL) measure to assess social support, only one study showed social support associated with both SBP and DBP dipping [51], one study associated social support with only SBP dipping [52], one study found no results for either SBP or DBP dipping [53], and one study [54] found a marginally significant association between social support and SBP dipping. Thus, one aim of our study was to examine the discrepancies between findings.

Social support can be broadly assessed by the ISEL, measuring perceptions of functional support including tangible, emotional, and informational support, and feelings of belonging. However, none of the studies noted above from Fortmann and Gallo’s meta-analysis examined the specific domains of the ISEL. Because different domains of social support can be more beneficial, depending on one’s needs, these specific types should be examined individually. Additionally, a significant portion of the literature has focused on hypertensive individuals, yet research shows that an increased blood pressure trajectory over time is associated with poor health outcomes even 25 years later [27]. Thus, it would be beneficial to understand the impact of social support on nocturnal dipping in a normotensive sample under 50 years of age, a point in life where individuals could make social changes that could decrease the risk of developing hypertension later in life.

In an effort to better understand this impact, we collected ABP on 179 normotensive individuals under 50 years of age over a 24-h period, and data on their social support. Because social support has been associated with stress, and stress can influence BP, we also looked at the association of stress on dipping. Additionally, based on recent work showing the prognostic value of nocturnal dipping at predicting cardiovascular disease over the prognostic value of 24-h blood pressure readings [35,36], and on the recent work on the association of social support on daytime BP [44], we looked at the impact of social support on daytime, nighttime, and 24-h ABP. Finally, we examined the effect of social support on nocturnal dipping using the full ISEL measure. Because social support can be seen in different facets as measured in the ISEL, we also examined the specific dimensions of social support. Additionally, we expected that this association would be moderated by sex based on the literature that has identified sex as an independent predictor of daytime and nocturnal BP and nocturnal BP dipping [55].

## 2. Materials and Methods

### 2.1. Criteria

Participants having the following conditions were excluded: medical conditions/medications with a cardiovascular component (e.g., hypertension or psychological problems for which they were being medically treated; see Cacioppo, Malarkey [56]) and a self-reported body mass index (BMI) no higher than 29.9, as 30 or higher is classified as obese, and hypertension and obesity are highly correlated. Participants were required to have a smartphone in order to complete a diary reading (see Measures below), at each BP reading. Each participant was given a personalized access code to the diary website.

### 2.2. Participants

In total, 179 participants (male, *n* = 91, 55%; female, *n* = 88, 45%) were recruited through a university, social media, and the community. All participants were over 21 and under 50 years of age, married, and currently living with their spouse. The mean age of participants was 24.85 years (SD = 4.10, range 21–46), and average length of their marriage was 2.99 years (SD = 2.04; range 1–18). Most were White (91.53%) and college educated (46.89% with college degree or higher; 51.98% currently pursuing a college degree), with 46.88% reporting an income over USD 30,000 (See Table 1).

### 2.3. Procedure

Following informed consent, eligible participants completed questionnaires related to perceptions of social support. Participants were then fitted with an ABP monitor and given detailed instructions on its use, shown how to stop a reading if needed (e.g., while driving, in a work meeting, etc.) and how to stop all readings if they chose to end the study early. Monitors were set to take a reading randomly twice an hour throughout the day and once per hour overnight. Participants were also given instructions on completing the diary entry (see Measures section below) and instructed to complete the entry within three to five minutes after the ABP monitor took a reading; diary entries were not required overnight. Participants returned the equipment the following day and received compensation. Participants were paid USD 75 each in cash.

### 2.4. Measures

#### 2.4.1. Physiological Measures

Ambulatory blood pressure was obtained using the Oscar 2 (Suntech Medical Instruments, Raleigh, NC, USA). The Oscar 2 was designed specifically for ABP assessment and has been validated for both SBP and DBP by international guidelines [57]. It utilizes codes that may signify problems with the estimation of ABP readings. Based on prior research [58], readings associated with weak Korotkoff sounds, measurement timeout, and air leaks were deleted. Outliers associated with artifactual readings identified using criteria by Marler, Jacob [59] were also discarded; these included: (a) SBP <70 mmHg or >250 mmHg, (b) DBP <45 mmHg or >150 mmHg, and (c) SBP/DBP < [1.065+ (0.00125 × DBP)] or >3.0.

#### 2.4.2. Psychological and Relationship Measures

The Perceived Stress Scale (PSS). The PSS is a ten-item assessment to measure stress perceptions and predict health-related outcomes associated with stress appraisal. This widely used assessment has been shown to have adequate psychometric properties and is related to other stress, health, and satisfaction measures [60]. Good reliability was demonstrated for the current study at 0.86.

Diary. Each participant completed a diary entry on their smartphone for each BP reading during the day. Piloting showed an entry took less than 2 min to complete. The diary collected information on standard control BP variables, and participants were instructed to complete the diary within 5 min following the BP reading. A time/day stamp allowed us to verify the diary entry was completed on time. Readings which were not completed within the 5 min window were discarded.

Sleep Quality. Sleep was assessed using a single item measure the following morning in which participants rated their sleep the previous night compared to an ordinary night on a 1–7 scale (1 = extremely bad; 7 = extremely good).

Interpersonal Support Evaluation List (ISEL). The ISEL [61] assesses network-level functional social support, measuring specific domains of appraisal, self-esteem, belonging, and tangible support. The ISEL has shown an overall internal consistency of 0.83. Our study demonstrated good reliability at 0.78.

### 2.5. Statistical Methodology

Data were analyzed using SAS version 9.4. Descriptive statistics were computed to examine demographics and baseline SBP and DBP in addition to average daily SBP and DBP, sleeping SBP, ISEL, and PSS averages. Mixed model (MIXED PROC) regressions were used to analyze associations between social support and nocturnal blood pressure dipping. Three steps were followed in running the regression models. The first step was to determine which covariates were significant predictors of the dependent variable (nocturnal dipping) by using forward selection methods. The second step was to run regression models. One regression model had PSS as the dependent variable, controlling for significant covariates from step one (age, BMI, posture, consumption of foods or drinks, and activity since the prior reading). The second regression model used ISEL as the dependent variable controlling for the same significant covariates. The last step was to conduct multiple group analysis to investigate gender differences in the models. Statistical significance was set at *p* < 0.05.

Nocturnal dipping was calculated as the change from daytime to nighttime BP. It has been measured by some researchers using the night–day ratio, with dipping (>0.8 and <0.9), extreme dipping (≤0.8), non-dipping (>0.9 and ≤1.0), and reverse dipping (>1.0). Using these criteria, among our sample, 16.77% would be classified as extreme dippers, 48.04% as dippers, 24.58% as non-dippers, and 10.61% as inverted dippers. Thus, it was more heavily distributed among dippers and non-dippers, and extremes in either direction were reasonably equivalent. We therefore treated nocturnal dipping dichotomously (dippers classified according to a dipping ratio of BP night/day; dippers were ≤0.90 and non-dippers were >0.90) taking the average of the daytime BP and the average of the night-time BP readings (time from self-reported bedtime to self-reported rising).

## 3. Results

### 3.1. Preliminary Analysis

The mean number of readings per participant was 36.98 (range 22–46) for the 24 h period. All outliers due to artifactual readings were discarded as noted above. Percentage of discarded readings per participant was 1.48% (M = 0.55, range 1–6). The SBP baseline average was 122 (SD = 12.19), and DBP baseline was 71.9 (SD = 7.84). Daily SBP average was 135.4 (SD = 18.83), and daily DBP average was 77 (SD = 9.92). Sleeping SBP average was 119.67 (SD = 20.82), and sleeping DBP average was 60.47 (SD = 10.26). ISEL scores ranged from 15–45, with a mean score of 36.39 (SD = 5.59). The PSS average was 16.5 (range 1–33; SD = 6.74), and sleep quality average was 4.06 (range 1–7; SD = 1.29) (Table 2).

We found social support associated with stress, such that those with greater perceived social support demonstrated less stress (B = −0.69, SE = 0.01, t(8141) = −62.55, *p* < 0.001). We next examined whether stress was associated with SBP or DBP dipping. Stress was associated with both SBP and DBP dipping and was thus included in the model.

We then examined daytime blood pressure readings and social support. Consistent with the Uchino findings, neither daytime SBP (B = −0.08, SE = 0.08, t(651) = −0.89, *p* = 0.37) nor DBP (B = 0.01, SE = 0.05, t(849) = 0.14, *p* = 0.37) was associated with social support. We then looked at nighttime readings and social support. SBP (B = 0.36, SE = 0.18, t(183) = 1.98, *p* = 0.04) was associated with social support, but DBP was not (B = 0.13, SE = 0.10, t(223) = 1.34, *p* = 0.18). Neither 24 h SBP (B = −0.07, SE = 0.08, t(677) = −0.89, *p* = 0.37) nor 24 h DBP (B = 0.00, SE = 0.04, (865) = 0.09, *p* = 0.92) was associated with social support.

### 3.2. Primary Analysis

Age, BMI, sex, position at time of reading, caffeine consumption, activity level, and sleep quality were significant predictors and were added to the model. We ran our first analysis on the full ISEL measure, capturing all domains in one score. As expected, nocturnal dipping was associated with perceptions of social support, such that those reporting low levels of social support showed blunted DBP dipping (B = −0.41, SD = 0.06, t(3801)= −7.24, *p* < 0.001). SBP dipping was not associated with social support (*p* = 0.118). We then looked at each domain separately to parse out the effect. Self-esteem was associated with both SBP dipping and DBP dipping (B = −0.64, SE = 0.09, t(3824) = −6.59, *p* < 0.001; B = −0.79, SE = 0.14, t(3801) = −5.74, *p* < 0.001), such that lower self-esteem support was associated with blunted dipping. Tangible support was associated with both SBP (B = −0.39, SE = 0.06, t(3824) = −6.20, *p* < 0.001) and DBP dipping (B = −1.75, SE = 0.19, t(3801) = −9.31, *p* < 001), such that lower tangible support was associated with blunted dipping. Belonging was associated with SBP (B = 0.36, SE = 0.05, t(3824) = 6.68, *p* < 0.001) and DBP (B = 0.38, SE = 0.11, t(3809) = 3.27, *p* = 0.001), such that less belonging support was associated with blunted dipping. Appraisal support was associated with blunted DBP (B = −1.68, SE = 0.19, t(3809) = −8.97, *p* < 0.001, but not with SBP (B = −0.07, SE = 0.05, t(3824) = −1.4, *p* = 0.16) (Table 3).

### 3.3. Effect of Gender

We examined whether the effects of social support on nocturnal dipping varied by sex. We found sex significantly interacted with dipping, such that women benefited more from total social support for SBP dipping (B = −0.298, SE = 0.034, t(3823) = −8.68, *p* < 0.001), but not DBP (B = 0.004, SE = 0.05, t(3800), *p* = 0.94). Looking at specific domains, women benefited more than men from tangible support for SBP (B = −0.55, SE = 0.097, t(3823) = −5.64, *p* < 0.001), but neither benefited for DBP (B−34, SE= 0.18, t(3800) = −1.86), *p* = 0.06). Women benefited more from belonging support for SBP (B = −0.64, SE = 0.12, t(3823) = −5.16, *p* < 0.001), but men benefited more for DBP (B = 0.88, SE = 0.13, t(3813) = 6.59, *p* < 0.001). Neither sex benefited from self-esteem support. Women benefited more than men from appraisal support for SBP (B = −1.154, SE = 0.09, t(3823) = −12.50, *p* < 0.001), but neither benefited for DBP (B = −0.07, SE = 0.05, t(3808) = −1.4, *p* = 0.161) (Table 4).

## 4. Discussion

Our main findings show social support associated with dipping, such that those individuals who perceive they have less social support demonstrate blunted nocturnal dipping for DBP but not for SBP. When we examined support by specific domains, we found both SBP dipping and DBP dipping associated with social support within the specific domains of tangible and self-esteem support. Further, we extended the prior literature showing the association between social support and health by examining normotensive individuals under 50 years of age. It is important to note that while overall social support was not associated with blunted SBP dipping, overall social support was associated with DBP dipping, and DBP carries its own risks separate from SBP. Whereas high SBP readings indicate an increase in the risk for heart disease such as heart attacks, heart failure, kidney disease, and overall mortality, high DBP is linked to higher risk of abdominal aortic aneurysm. The American Heart Association notes that there is an emphasis on SBP, yet research has shown that each increase of 10 mmHg in DBP is associated with a 28% risk of developing an abdominal aortic aneurysm [62]. It is therefore important to take both SBP and DBP into account when assessing risk.

While we found DBP dipping to be associated with social support, we also found the same results as Uchino and colleagues on daily blood pressure and social support, such that daytime SBP and DBP were not associated with social support. We also found that nighttime DBP was not associated with social support, nor did we find an association for either 24 h SBP or DBP, although nighttime SBP was associated with social support. This is interesting, as both daytime and nighttime ABP are used to determine nocturnal dipping. This seems to suggest that it is the combination of daytime and nighttime blood pressure, specifically using the ratio of daytime and nighttime blood pressure, that is more useful as a measure of cardiovascular disease risk than using daytime or nighttime measures alone. The lack of association between 24 h SBP or DBP and social support is also indicative of the benefits of using nocturnal dipping as a health outcome, rather than BP alone, whether 24 h, daytime, or nighttime.

Social support is multidimensional and can influence health through various pathways. We gained a better picture of the contributions of social support to nocturnal dipping when we divided social support into its component parts. Tangible support predicted both SBP and DBP dipping. It can be expected that tangible support is an aspect of social support that was associated with both SBP and DBP dipping. Tangible support is one specific support that individuals find most beneficial. It can include provisions of shelter, food, or financial help, and the perception of the availability of such assistance when needed can significantly reduce stress. Financial stress can be a particular source of distress and is related to poor psychological and physiological health for those with low levels of perceived tangible support, with a six- to seven-fold increased odds ratio for poor psychological well-being and psychosomatic symptoms [63]. While there is a large body of literature on the benefits of tangible support on health outcomes, there is little addressing tangible support and nocturnal dipping. Our findings address this gap and demonstrate that in addition to contributing to other health-related outcomes, tangible support impacts nocturnal dipping.

SBP dipping and DBP dipping were also associated with self-esteem social support. Self-esteem social support assessments include items such as “Most people I know think highly of me.” Such support can be beneficial in terms of feeling valued by others. Self-esteem support can increase one’s ability to ask for help when help is needed, as it may help decrease feelings of burdensomeness. Thus, being able to ask for needed support could be manifest in healthy nocturnal dipping, as shown in this study.

These findings indicate the importance of examining social support within the differing domains, as one kind of support may be more effective at reducing stress than another. This is not to say that of benefits generally, only these two aspects of support (tangible and self-esteem) are beneficial, and the other aspects are not. Rather, our findings indicate that normotensive individuals under the age of 50 may benefit more from these specific types of support than older normotensive or hypertensive individuals. This is important to our understanding of the link between social support and health, as the benefits of social support have been shown to be more effective if they are applicable to the needs of the individual. In other words, one who needs emotional support in a time of stress will not find informational support helpful. Support is most effective if tailored to the specific needs of the individual.

It is expected that women would benefit more from social support. Traditionally, women offer more social support to others and when facing stress, tend to give and receive more social support than males. Part of this phenomenon, as noted in the literature, may be because men are more likely to react to stress with a “fight or flight” orientation, while women tend to use the “tend-and-befriend” response, with ‘befriend’ referring to creating and maintaining social networks. Men have more diverse social ties, but also more negative interactions. Women’s networks tend to be more homophilic and homogeneous and family-based than men’s, which may make it easier for a woman to ask for help from someone similar to her. Further, women are more likely to offer support to their same-sex friends than men are to offer support to their same-sex friends, and men are less likely than women to seek or provide support. These differences in social network structure and function may lead to women having a larger network from which to draw social support, be more likely to ask for help, and thus more likely to benefit from their social support.

Women’s SBP dipping and DBP dipping were also associated with belonging and appraisal social support. Belonging assessments include items such as “If I decide one afternoon to go to a movie, I could easily find someone to go with me.” Such belonging support can be beneficial in terms of enhancing mood and feeling a sense of acceptance and belonging by others. This sense of acceptance and belonging can help individuals to cope with stressors more effectively and could help one to avoid certain stressors to begin with. Appraisal support is measured with items such as “There is someone I can turn to for advice about handling problems with my family.” This type of advice or guidance can be a particularly effective type of support for women as demonstrated in the healthier dipping profile seen in our participants.

The current study is a novel contribution to the literature, as we have examined social support and health in a relatively under-investigated health outcome, which is a known contributor to cardiovascular disease. Further, we examined those at greater risk of loneliness and lack of social support and have done so by looking at the overall ISEL measure and the individual domains of social support. Lastly, we have specifically focused on a normotensive sample, which is likely more representative of those in the population of individuals under 50 years of age.

### Limitations and Future Work

While these findings are important, certain limitations apply. Our sample was predominately White and educated, and all were heterosexual and married; thus, it is not clear how these findings would relate to unmarried individuals, people of color, or non-heterosexual individuals. High BP is also more common is non-Hispanic Black adults than in non-Hispanic White adults and is more common in African Americans than in other ethnic groups. It would then be important to look at nocturnal BP in a more diverse sample. We used BMI as a criterion for exclusion based on the large amount of prior research that has used this measure. BMI is also a quick way to determine qualification before participants are scheduled to come to the lab. However, the body adiposity index (or hip-to-waist ratio) is now generally the preferred method and may have more accurately assessed obesity. We did not use the body adiposity index, as it could not be assessed until the participant arrived at the lab. Decisions on eligibility needed to be assessed earlier in the screening process. Additionally, we did not assess whether participants had taken a nap during the day, which could impact sleep time/quality. Our study was also cross-sectional; thus, the social support needs of the individual at this specific point may have influenced their response to the individual types of support (e.g., tangible vs. emotional). We also measured nocturnal blood pressure over a single 24 h period. It would be beneficial to measure over several 24 h periods. Finally, it is important to note that while ABP is a valuable and well-validated tool for measuring daytime and nighttime BP and has consistently found associations between cardiovascular measures and social support, the recent meta-analysis on daytime ABP found no such connection [44]. Our findings are consistent with these findings, and yet we showed social support associated with total DBP dipping, and with both SBP and DBP dipping within the various component parts of the ISEL measure. It is therefore important that future research examine how the findings of this study and from current research fit into the overall literature on social support and cardiovascular health.

## 5. Conclusions

Despite these limitations, our study demonstrated the importance of social support for normotensive individuals who may be at greater risk for insufficient support, and the importance of examining social support more broadly. Further, our study demonstrated the importance of examining nocturnal blood pressure in addition to using daytime or overnight blood pressure to assess the benefits of social support on cardiovascular health. Future studies should also consider an examination of nocturnal dipping in a more diverse sample.

## Figures and Tables

**Table 1 ijerph-20-04579-t001:** Demographic characteristics.

Variable	Mean (SD)	Min	Max	N (%)
Age (Years)	24.85 (4.10)	21	46	179 (100.00)
Marriage length (Years)	2.99 (2.04)	1	18	179 (100.00)
Sex				
Male				88 (49.00)
Female				91 (51.00)
Income (USD)				
Less than 15,000				34 (19.20)
15,000–29,999				61 (33.90)
30,000–49,000				47 (26.50)
50,000–69,000				19 (10.17)
70,000 or more				17 (9.60)
Prefer not to answer				1 (0.60)
Ethnicity				
White				64 (91.53)
Hispanic/Latino				10 (5.65)
Asian				2 (1.13)
Native American				1 (0.56)
Pacific Islander				1 (0.56)
Other				1 (0.56)
Education				
Graduated from high school				2 (1.13)
Had partial college				92 (51.98)
Graduated from college				57 (32.20)
Had partial graduate/professional school				16 (9.04)
Graduated from graduate/professional school				10 (5.65)

**Table 2 ijerph-20-04579-t002:** Characterization of participants’ health status.

Variable	Mean (SD)	Min/Max Baseline Mean (SD)	Day Mean (SD)	Night Mean (SD)
ISEL	36.39 (5.59)	15/45		
PSS	16.50 (6.74)	1/33		
Sleep quality	4.06 (1.29)	1/7		
SBP		93/161	122.00 (12.19)	135.40 (18.83) 119.67 (20.82)
DBP		54/96	71.90 (7.84)	77.00 (9.92) 60.47 (10.26)

Note: ISEL = social support; PSS = stress perception; SBP = systolic blood pressure; DBP = diastolic blood pressure.

**Table 3 ijerph-20-04579-t003:** Regression models predicting social support and stress perception overall.

Variable	*B*	SE	t	*p*-Value
PSS				
ISEL Total	−0.690	0.010	−62.55	<0.001
PSS				
SBP	−0.014	0.001	−9.55	<0.001
DBP	−0.020	0.001	−17.34	<0.001
ISEL DAYTIME				
SBP	−0.075	0.084	0.89	0.372
DBP	0.007	0.053	−0.14	0.891
ISEL NIGHTTIME				
SBP	0.362	0.183	1.98	0.049
DBP	0.136	0.109	1.34	0.182
ISEL 24-h				
SBP	0.072	0.081	−0.89	0.372
DBP	0.004	0.048	0.09	0.927
ISEL DIPPING Total				
SBP	−0.039	0.025	−1.56	0.118
DBP	−0.413	0.060	−7.24	<0.001
ISEL DIPPING Tangible Support				
SBP	−0.390	0.060	−6.20	<0.001
DBP	−1.750	0.190	−9.31	<0.001
ISEL DIPPING Belonging				
SBP	0.360	0.050	6.68	<0.001
DBP	0.380	0.110	3.27	<0.001
ISEL DIPPING Self-Esteem				
SBP	−0.640	0.090	−6.59	<0.001
DBP	−0.790	0.138	−5.74	<0.001
ISEL DIPPING Appraisal				
SBP	−0.070	0.050	−1.40	0.160
DBP	−1.680	0.190	−8.97	<0.001

Note: ISEL = social support; PSS = stress perception; SBP = systolic blood pressure; DBP = diastolic blood pressure.

**Table 4 ijerph-20-04579-t004:** Regression models in predicting social support by gender.

Variable	*B*	SE	t	*p*-Value
ISEL Total				
SBP	−0.290	0.030	−0.868	<0.001
DBP	0.004	0.050	0.080	0.940
ISEL Tangible Support				
SBP	−0.550	0.090	−5.640	<0.001
DBP	−0.340	0.180	−1.860	0.060
ISEL Belonging				
SBP	−0.640	0.012	−5.16	<0.001
DBP	0.880	0.130	6.59	<0.001
ISEL Self-Esteem				
SBP	0.040	0.140	0.31	0.750
DBP	−0.050	0.260	−0.19	0.850
ISEL Appraisal				
SBP	−1.150	0.090	12.50	<0.001
DBP	−0.070	0.050	−1.40	0.160

Note: ISEL = social support; PSS = stress perception; SBP = systolic blood pressure; DBP = diastolic blood pressure.

## Data Availability

Data are available from authors upon request and IRB approval.

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
