# Peer review of "Social Support: The Effect on Nocturnal Blood Pressure Dipping"

_ijerph, 2023, doi:10.3390/ijerph20054579_

Round 1

Reviewer 1 Report

This study aims to investigate the impact of social support on nocturnal dipping in the normotensive population. The article is well-written, and all the key findings are well-explained in the discussion section. Here are a few minor suggestions.

1. Line 36 “A lack of social support has been associated with a higher incidence of cancer [16]” is unclear. Please explain in 1-2 sentences how social support has been associated with higher cancer incidence. Explain distress or any other underlying mechanism in a few sentences.

2.     In the participant section, please include how many men and women were included in the study.

3.     Did you measure any blood biomarkers/ hormones (Oxytocin, cortisol, epinephrine, etc.)? 

Author Response

Thank you for your review and suggestions/recommendations.

Reviewer 1 asked that we explain in 1-2 sentences how social support has been associated with higher cancer incidence. We appreciate the opportunity to clarify this association. We have now inserted more information on cancer incidence and support and  have used more updated references(per Reviewer 2’s request that we update our references)

Reviewer 1 also asked that we list the sex of the participants. We have now inserted those values. 91 were male (55%) and 88 were female (45%).

Reviewer 1 asked if we had measured other biomarkers such as oxytocin, cortisol or epinephrine. We did not collect these biomarkers.

Reviewer 2 Report

While reviewing the paper by Birmingham et al. "Social Support: The Effect on Nocturnal Blood Pressure Dipping" I had mixed feelings. On the one hand, the authors tried to obtain new scientific facts by examining healthy individuals under 50 years of age, and studying their level of social support, as well as nocturnal blood pressure dipping. In the course of the study, they showed a greater level of insufficient support with nocturnal blood pressure non-dipping. On the other hand, the quality of the study and the format of its presentation in the manuscript contain fundamental shortcomings that significantly reduce the value of the results obtained.

1. In the Introduction section, the authors cite in detail well-known literary sources of past years (61 out of 85 sources more than 10 years old, 29 sources published in the last century). However, the authors do not consider publications on the research topic, although there are not only individual publications (1,2), but even meta-analyses (3,4) on this issue. Therefore, the relevance and novelty of the study raises reasonable doubts.

2. It looks redundant to use references to 70 sources in the Introduction.

3. I do not see the need to formulate 3 hypotheses of the study, it was quite possible to reflect all the intentions of the authors in a clearly formulated goal of the study.

4. The Material and Methods section is poorly written - part of the information for this section is presented in other sections (Introduction, Results).

5. In section 2.5. Statistical Model does not represent all the methods of statistical analysis used by the authors.

6. In the Results section, data are presented only in the text, not a single table is given, which makes it extremely difficult for the authors to provide information. There is a link to table 1, but it is not presented either.

7. The Discussion section begins with a literature reference, which is more appropriate in the Introduction section.

8. In the Discussion section, the authors write that "Our main findings indicate social support associated with dipping such that those individuals who perceive they have less social support demonstrate blunted nocturnal dipping for DBP but we did not find this effect on SBP dipping". However, they do not confirm the novelty of these data anywhere in the manuscript, which they usually do when comparing their results with previous studies.

References:

1.       Rodriguez CJ, Burg MM, Meng J, Pickering TG, Jin Z, Sacco RL, Boden-Albala B, Homma S, Di Tullio MR. Effect of social support on nocturnal blood pressure dipping. Psychosom Med. 2008 Jan;70(1):7-12. doi: 10.1097/PSY.0b013e31815aab4e.

2.       Spruill TM, Shallcross AJ, Ogedegbe G, Chaplin WF, Butler M, Palfrey A, Shimbo D, Muntner P, Sims M, Sarpong DF, Agyemang C, Ravenell J. Psychosocial Correlates of Nocturnal Blood Pressure Dipping in African Americans: The Jackson Heart Study. Am J Hypertens. 2016 Aug;29(8):904-12. doi: 10.1093/ajh/hpw008.

3.       Uchino BN, Baucom BRW, Landvatter J, de Grey RGK, Tacana T, Flores M, Ruiz JM. Perceived social support and ambulatory blood pressure during daily life: a meta-analysis. J Behav Med. 2022 Aug;45(4):509-517. doi: 10.1007/s10865-021-00273-3.

Fortmann AL, Gallo LC. Social support and nocturnal blood pressure dipping: a systematic review. Am J Hypertens. 2013 Mar;26(3):302-10. doi: 10.1093/ajh/hps041. 

Author Response

Thank you for your suggestions/recommendations. 

Reviewer 2

In terms of references, Reviewer 2 noted: “the authors cite in detail well-known literary sources of past years (61 out of 85 sources more than 10 years old, 29 sources published in the last century). However, the authors do not consider publications on the research topic, although there are not only individual publications (1,2), but even meta-analyses (3,4) on this issue. Therefore, the relevance and novelty of the study raises reasonable doubts.”

“The Discussion section begins with a literature reference, which is more appropriate in the Introduction section.”

“It looks redundant to use references to 70 sources in the Introduction”

We appreciate your request to update our references and agree that it will make the manuscript more relevant. We have left some of the more seminal studies in the manuscript, and updated others. We have also reduced the number of references in the introduction and removed the references from the beginning of the discussion section, and  also eliminated the first portion of the Discussion section.

“I do not see the need to formulate 3 hypotheses of the study, it was quite possible to reflect all the intentions of the authors in a clearly formulated goal of the study.”

Thank you. We have now revised this section.

"The Material and Methods section is poorly written - part of the information for this section is presented in other sections (Introduction, Results).”

We have rewritten portions of the section to eliminate any redundancy, and rearranged portions which may better fit elsewhere. We hope this makes the section clearer.

"The Statistical Model does not represent all the methods of statistical analysis used."

We agree that the manuscript would be better served to include this. We have now added a fuller explanation of the method of analysis used.

"In the results section, data are presented only in the text."

We apologize that these tables were not included in the manuscript. We agree that they would help the reader disseminate the findings. We have now included 4 tables which represent the demographics and findings.

"The Discussion section begins with a literature reference, which is more appropriate in the Introduction section."

Thank you. We have now removed this portion.

"In the Discussion section, the authors write that "Our main findings indicate social support associated with dipping such that those individuals who perceive they have less social support demonstrate blunted nocturnal dipping for DBP but we did not find this effect on SBP dipping". However, they do not confirm the novelty of these data anywhere in the manuscript, which they usually do when comparing their results with previous studies."

Thank you for this important observation. It has indeed been left out, and we have now rewritten portions of the manuscript to reflect the importance of our finding.

Reviewer 2 also suggests we incorporate specific studies and meta-analyses to further make our case for the necessity of the study.

We have now included  several studies you have suggested.

Round 2

Reviewer 1 Report

No further comments. 

Author Response

Thank you. 

Reviewer 2 Report

The authors significantly corrected the manuscript and answered my comments and questions. However, not all answers satisfied me completely.

1. The Introduction section still contains a lot of well-known, long-published information. The authors have somewhat reduced and corrected the references, but this is clearly not enough. As before, more than half of the references to sources older than 10 years, 13 - to studies of the last century. One gets the impression that the authors have been studying this problem for a long time and simply use links from past publications. In any case, such old references do not agree well with the evidence of the relevance of the study. Yes, the authors added one relatively old systematic review (Fortmann AL and Gallo LC), but ignored a recent meta-analysis on the topic of the manuscript (Uchino BN). It is all the more strange that the first author of the manuscript has joint publications with the authors of this meta-analysis. I think that this section needs to be improved, as well as the corresponding references.

Author Response

Reviewer 2: Thank you for your request for updated references. We have now revised the Introduction to include references which are generally within the last 5-8 years, most within the last 3.   We've also included the Uchino paper (and agree that this is an important paper and should be included) and discussed the implications of the meta-analysis for our study.  We completed additional analysis to better address these implications, and added those findings to the manuscript and the tables. We've kept track changes to show the new papers in the reference list, and also track changed the citations in the manuscript, for ease of finding. We agree that updating these references could, and has, made the paper better. We hope these updates and additions will make this paper suitable for publication.